# Hyperhomocysteinemia in Adult Patients: A Treatable Metabolic Condition

**DOI:** 10.3390/nu16010135

**Published:** 2023-12-30

**Authors:** Domingo González-Lamuño, Francisco Jesús Arrieta-Blanco, Elena Dios Fuentes, María Teresa Forga-Visa, Monstserrat Morales-Conejo, Luis Peña-Quintana, Isidro Vitoria-Miñana

**Affiliations:** 1Pediatrics Department, Marqués de Valdecilla University Hospital, 39008 Santander, Spain; 2Adult Inborn Errors of Metabolism Unit, Ramón y Cajal University Hospital, 28034 Madrid, Spain; arri68@hotmail.com; 3Endocrinology and Nutrition Department, Virgen del Rocío University Hospital, 41013 Sevilla, Spain; elenadiosfuentes@gmail.com; 4Endocrinology and Nutrition Department, Clínic Hospital, 08036 Barcelona, Spain; mtforga@clinic.cat; 5Unit for Congenital Metabolic Diseases and Other Rare Diseases, Internal Medicine Department, 12 de Octubre University Hospital, 28041 Madrid, Spain; moralmon@yahoo.es; 6Pediatric Gastroenterology and Nutrition Unit, Insular Materno-Infantil University Hospital Complex, Asociación Canaria de Investigación Pediátrica, Centro de Investigación Biomédica en Red de la Fisiopatología de la Obesidad y Nutrición, University of Las Palmas de Gran Canaria, 35016 Las Palmas de Gran Canaria, Spain; luis.pena@ulpgc.es; 7Nutrition and Metabolic Diseases Unit, La Fe University Hospital, 46026 Valencia, Spain

**Keywords:** homocysteine, hyperhomocysteinemia, homocystinuria, inborn metabolic diseases, transsulfuration, remethylation, thrombotic events, Marfanoid habitus, ectopia lentis, betaine

## Abstract

Hyperhomocysteinemia (HHcy) is recognized as an independent risk factor for various significant medical conditions, yet controversy persists around its assessment and management. The diagnosis of disorders afffecting homocysteine (Hcy) metabolism faces delays due to insufficient awareness of its clinical presentation and unique biochemical characteristics. In cases of arterial or venous thrombotic vascular events, particularly with other comorbidities, it is crucial to consider moderate to severe HHcy. A nutritional approach to HHcy management involves implementing dietary strategies and targeted supplementation, emphasizing key nutrients like vitamin B6, B12, and folate that are crucial for Hcy conversion. Adequate intake of these vitamins, along with betaine supplementation, supports Hcy remethylation. Lifestyle modifications, such as smoking cessation and regular physical activity, complement the nutritional approach to enhance Hcy metabolism. For individuals with HHcy, maintaining a plasma Hcy concentration below 50 μmol/L consistently is vital to lowering the risk of vascular events. Collaboration with healthcare professionals and dietitians is essential for developing personalized dietary plans addressing the specific needs and underlying health conditions. This integrated approach aims to optimize metabolic processes and reduce the associated health risks.

## 1. Introduction

Hyperhomocysteinemia (HHcy) is a metabolic condition characterized by elevated blood homocysteine (Hcy) levels, which is implicated in various disorders, serving as a potential risk factor for serious complications [1,2,3]. Despite its recognition, the assessment and management of HHcy remain contentious, marked by the conflicting results of studies evaluating its impact on reducing cardiovascular and cerebrovascular disease risks.

Hcy, a non-essential amino acid derived from methionine (Met) metabolism, participates in a complex cycle involving Met and cysteine (Cy) through a transsulfuration mechanism [4]. Met, primarily obtained from dietary proteins, undergoes recycling into Hcy via a methyl group donation reaction. The remethylation of Hcy to Met, facilitated by methionine synthase (MS), requires folate derivatives and vitamin B12. The transsulfuration pathway, with vitamin B6 as a cofactor, converts Hcy into Cy and subsequently into sulfate (Figure 1).

Enzymatic deficiencies in these pathways can lead to HHcy, with elevated Hcy levels indicating metabolic dysfunction and predisposing individuals to arterial and venous thromboembolism by damaging vascular endothelial cells [5]. While some debate the direct link between HHcy and thrombosis risk [6,7], it is well established that lowering the Hcy levels reduces the cardiovascular risk in classic homocystinuria (HCU) patients, afflicted with a severe deficiency of the CβS enzyme [8,9].

Classic HCU patients, marked by CβS enzyme deficiency, often encounter thrombotic or atherosclerotic diseases early in life, alongside neurological, psychiatric, and skeletal complications. The reduction in the Hcy levels has demonstrated efficacy in slowing the processes related to brain atrophy [10]. However, a 2017 Cochrane review [3] suggests that therapies targeting mild forms of HHcy may not significantly impact stroke prevention and minimally affect coronary heart disease prevention; they focused on mild elevations commonly encountered in clinical practice.

Despite the analytical challenges associated with Hcy measurement, disorders in Hcy metabolism manifest through simple assessments of plasma Hcy concentrations. Marked elevations are observed in homozygous CβS deficiency, while more moderate increases occur in heterozygous CβS deficiency and folic acid metabolism disorders like methylenetetrahydrofolate reductase (MTHFR) deficiency. Additional metabolic studies, encompassing amino acids, plasma vitamins, and organic acids in urine, may be necessary for a comprehensive evaluation [5].

The substantial delay in detecting HHcy, coupled with the absence of standardized therapeutic protocols and a proper etiological diagnosis, complicates the management of this condition. This article underscores the need to investigate HHcy in various clinical contexts and advocates for fundamental metabolic studies to identify the mechanisms involved in moderate or severe forms, guiding appropriate treatment.

## 2. Types of Hyperhomocysteinemias (HHcy)

HHcy, characterized by elevated blood Hcy levels exceeding 15 μmol/L, is influenced by a series of reactions involving essential vitamins (B6, B12, and folates) and enzymes, notably MTHFR. Nutritional deficiencies, including alcohol consumption, untreated celiac disease, and prolonged use of proton pump inhibitors, can lead to elevated Hcy levels due to their crucial role in Hcy metabolism [11,12]. Elevated Hcy is also associated with various conditions such as cognitive impairment, chronic kidney disease, hypothyroidism, psychiatric disorders, and bone mineralization disorders [13,14,15,16]. Inborn errors of Hcy metabolism (IEM) result in severe HHcy, categorized into mild (16 to 30 μmol/L), moderate (31 to 100 μmol/L), and severe (more than 100 μmol/L) forms, based on clinical impact [17] (Figure 2).

Severe forms, including classic HCU, stem from CβS enzyme deficiency, crucial for Hcy transsulfuration, accompanied by hypomethioninemia. Other IEMs leading to severe HHcy involve remethylation disorders, often due to MTHFR enzyme deficiency or defects in enzymes related to vitamin B12. Mutations in the MTHFR gene disrupt folate acid metabolism, resulting in methyltetrahydrofolate deficiency and HHcy with hypomethioninemia. Disorders related to cobalamin (cbl-) metabolism, particularly cblC, combine methylmalonic acidemia (MMA) with HCU [4,18]. These disorders manifest with significantly elevated Hcy levels (>50 μmol/L).

Acquired HHcy may result from lifestyle factors, clinical conditions, and medications. Renal insufficiency and certain medications, including folate antagonists, carbamazepine, nitric oxide, methotrexate, cholestyramine, and niacin, contribute to acquired HHcy [4,10,19]. Acquired forms may remain asymptomatic until the third or fourth decade of life, manifesting as thrombotic events. The recognition of these diverse types of HHcy highlights the need for comprehensive investigation, considering both congenital and acquired factors, to guide appropriate therapeutic interventions [20]. See Table 1.

## 3. Role of Homocysteine in Disease Processes (Toxicity)

The involvement of Hcy in disease processes, particularly its toxic effects, can be categorized into those directly attributed to Hcy and those stemming from secondary modifications in related metabolites and processes due to elevated Hcy levels. Although the precise pathophysiological effects of Hcy remain incompletely understood, lacking a unifying concept, its toxicity is undeniably linked to the unique reactivity of the Hcy SH group. One proposed mechanism is Hcy oxidation, generating superoxide anion radicals and hydrogen peroxide, leading to increased oxidative stress [22].

Additionally, posttranslational incorporation of Hcy thiolactone into proteins through N-homocysteinylation has been identified [23].

Elevated Hcy levels cause secondary disturbances, such as increased AdoHcy, leading to the inhibition of essential transmethylation reactions, including DNA methylation. Disturbances in folate coenzyme homeostasis and the deficiency of essential compounds contribute to the pathogenesis.

The association between HHcy and diseases, particularly vascular disease, has led to extensive research on the potential mechanisms, including alterations to blood vessel architecture, endothelial damage, oxidative stress, and inflammatory responses [24].

Hydrogen sulfide (H2S) that results from the transsulfuration process has been implicated in cardiovascular protection through redox balance and vessel relaxation. The paper by Wijerathne et al. (2020) emphasizes the role of the endogenous production of H2S in kidney ischemia–reperfusion injury and oxidative stress in the heart [25].

The neurotoxic effects of Hcy involve oxidative stress, DNA damage, protein thiolation, and homocysteinylation, triggering apoptosis and excitotoxicity. Inflammation during HHcy is associated with increased cytokines and changes in DNA methylation.

Hcy concentration is regulated by remethylation or transsulfuration pathways, with H2S implicated in neuroprotection. Reduced H2S levels under HHcy conditions may contribute to homocysteine-induced neurotoxicity.

Oxidative stress, endoplasmic reticulum stress, inflammation, and epigenetic modifications are suggested mechanisms of HHcy-induced blood–retinal barrier dysfunction. Brain inflammation in HHcy is linked to blood–brain barrier dysfunction and Alzheimer’s disease pathogenesis. The focus of Tawfik et al. emphasizes the effects of HHcy on the blood–retinal barrier and the controversial role in aging neurological diseases, highlighting potential mechanisms [26].

Finally, the process of normal bone formation is dependent on the collagen matrix and requires the synthesis of collagen crosslinks. The sulfhydryl group (–SH) of Hcy interferes with the formation of these crosslink precursors of collagen [27] and thus the normal synthesis of collagen and bone formation [28]. Patients with HHcy and HCU develop osteoporosis at a higher rate than normohomocysteinemic subjects [29].

HHcy is also known to cause an imbalance between matrix metalloproteinases (MMPs) and tissue inhibitors of metalloproteinases (TIMPs), leading to the accumulation of collagen in the aorta and resulting in stiffness and the development of hypertension [30]. HHcy plays a critical role in the development of various aortic diseases [31,32,33,34].

Although the exact mechanism of extracellular matrix (ECM) remodeling is unclear, emerging evidence implicates epigenetic regulation involving DNA methylation. Epigenetic mechanisms such as DNA methylation are known to control the expression of ECM components [35]. Although various studies report an aberrant DNA methylation pattern in the early stages of atherosclerosis [36] and aortic aneurysm [37], the role of DNA hypermethylation in aortic remodeling and arterial hypertension in HHcy remains unclear. Elevated Hcy levels contribute to various pathologies with emerging insights into epigenetic mechanisms. HHcy is associated with dysregulated DNA methylation, impacting gene expression in conditions such as vascular disease and neurodegenerative disorders. Alterations in histone modifications further contribute to inflammatory responses and endothelial dysfunction linked to HHcy. Dysregulation of microRNAs, small RNA molecules regulating gene expression, adds another layer to the pathogenesis of the cardiovascular and neurodegenerative diseases associated with HHcy. The intricate crosstalk between different epigenetic mechanisms reveals a complex regulatory network influenced by HHcy. Understanding these epigenetic aspects provides potential therapeutic targets, offering avenues for targeted interventions to mitigate the pathological consequences of elevated homocysteine levels [38].

## 4. Clinical Manifestations of HHcy

Disorders affecting Hcy metabolism present a diverse clinical spectrum contingent upon the enzymatic impairment’s nature and extent. Moderate to severe HHcy leads to significant morbidity and mortality, affecting multiple organ systems. In the IEM of Hcy, symptoms often emerge in early childhood, resulting in neurodevelopmental, skeletal, ocular, and vascular complications.

Classic HCU, a severe form of HHcy due to CβS deficiency, manifests with skeletal anomalies, ocular issues, neuropsychiatric concerns, and vascular problems. Without timely intervention, it can lead to psychiatric disorders, behavioral issues, and preventable intellectual disability. Recurrent severe vascular occlusions are a principal complication of HHcy, with even a moderate elevation in blood Hcy contributing independently to vascular diseases. Hcy serves as an atherogenic risk predictor and potent prognostic indicator for adverse cardiovascular events [19,20,21,39], emphasizing the clinical significance of HHcy and the importance of interventions to mitigate the risks and improve the outcomes [18]. Clinical suspicion arises with moderately elevated (31–100 μmol/L) or severe (>100 μmol/L) Hcy levels, and a definitive diagnosis involves identifying mutations in the *CβS* gene [40].

Other causes of HHcy include disorders affecting Hcy remethylation and sulfur amino acid metabolism. Vitamin B12 metabolism abnormalities, folate metabolism issues (*MTHFR* gene mutations), and abnormalities in MS metabolism contribute to HHcy. CblC deficiency exhibits neurological, ocular, and renal symptoms. MTHFR deficiency manifests with progressive encephalopathy, epilepsy, and psychiatric disorders. MS deficiency leads to megaloblastic anemia, intellectual disability, and psychiatric disorders. Understanding the varied clinical manifestations of HHcy and its underlying causes is crucial for an effective diagnosis and intervention.

## 5. Specific Organ or System Manifestations (Table 2)

### 5.1. Vascular

Elevated blood Hcy levels are associated with atherogenic and prothrombotic properties, leading to specific histopathological features in vascular lesions, including intimal thickening, elastic lamina rupture, smooth muscle hypertrophy, platelet accumulation, and occlusive thrombi formation. See Table 2.

**Table 2 nutrients-16-00135-t002:** Specific organ or system manifestations of HHCy.

HCU and Hematological/Vascular Pathology	HCU and Ophthalmological Pathology
Classical HCU - Early arteriosclerosis - Thromboembolism - Pulmonary - Cerebral - Myocardial infarction Remethylation Defects - Macrocytosis or megaloblastic anemia, neutropenia, or pancytopenia - Thromboembolic event	Classical HCU Inferior lens subluxation. Here 90% of patients progressively develop lens ectopia Severe myopia Cataracts > 15 years Remethylation Defects Maculopathy Retinitis pigmentosa and optic atrophy Central retinal vein occlusion (CRVO) in MTHFR deficiency
HCU and Neuropsychiatric Pathology	HCU and Fertility
Classical HCU - Intellectual disability - Mood disorders, anxiety, and obsessive–compulsive disorder - Psychotic symptoms may be present in the absence of any other symptom. - Visual hallucinations, agitation, and poor response to antipsychotics - Stroke (especially carotid thrombosis) Remethylation Defects - Schizophrenia and early-onset bipolar disorder - Acute psychiatric disorder - Leukoencephalopathy	HCU due to Remethylation Defects Infertility, especially in the case of recurrent miscarriages Preeclampsia, placental infarction, and placental abruption Intrauterine growth restriction Coagulation and platelet aggregation disorders Risk of cervical cancer Neural tube defects (NTDs) HCU due to CβS Deficiency Venous thrombosis Hypercoagulability Vascular damage with infarctions, thrombosis, and premature villous aging
HCU and Renal Disease	Others (Skeletal, Hearing Loss, …)
Hyperhomocysteinemia - Chronic kidney failure - Renal infarction Remethylation Defects - Thrombotic microangiopathy (TMA) - Hemolytic–uremic syndrome - Renal infarction - Proteinuria - Hypertension - Chronic renal failure	Skeletal Marfanoid habitus Osteoporosis Auditive Unilateral hearing loss (cblC) Increased susceptibility to noise-induced hearing loss

Studies report increased odds ratios (2.5 to 3.0) for venous thromboembolic disease (VTE) in individuals with Hcy levels exceeding two standard deviations above the normal range. Mild HHcy (16 to 30 μmol/L) may serve as a risk marker for recurrent VTE, although with some reservations.

At the arterial level, premature atherosclerosis and embolic complications affecting large and small vessels can manifest at any age. Arterial or venous thrombotic events, especially cerebral, occur when the total Hcy concentration exceeds 100 μmol/L. Without treatment, around 50% of patients with severe genetically induced HHcy experience vascular complications before age 30. In hereditary HCU, untreated individuals face early arteriosclerosis and a heightened risk of vascular complications. While these complications may arise in childhood, they more commonly present as initial signs during adulthood in approximately 10% of cases [41]. Up to 30% of HCU patients experience vascular events before turning 20, with half involving peripheral venous system thromboembolisms, 25% pulmonary embolisms, up to 33% cerebrovascular accidents, 11% peripheral arterial thromboembolisms, and 4% myocardial infarctions [42]. Appropriate treatment significantly reduces the risk of these events [43].

Consideration of an HCU diagnosis is warranted in individuals under age 55 experiencing venous or arterial thromboembolic events, particularly if recurrent. A strong suspicion arises when such events are accompanied by psychiatric symptoms, mild cognitive impairment, a Marfanoid appearance, or early lens dislocation history. Triggering situations, such as physical stress, surgery, the postpartum period, or prolonged fasting, should be considered. The absence or minimal relevance of other clinical manifestations may suggest milder phenotypes or responsiveness to vitamin B6 treatment [44].

### 5.2. Hematological

For HHcys resulting from remethylation disorders, hematological complications (megaloblastic anemia, neutropenia, and/or pancytopenia) are more prevalent than vascular complications. This association between hematological abnormalities and thromboembolic processes is more frequently observed in adult-onset presentations. These hematological abnormalities should be scrutinized, particularly in patients who also display visual deficits, retinopathy, peripheral neuropathy, ataxia, spinal degeneration, mild to moderate cognitive impairment, or psychiatric or behavioral disorders. See Table 2.

### 5.3. Marfanoid Habit and Osteoporosis

Skeletal involvement is marked by a Marfanoid phenotype and early osteoporosis. Elevated Hcy levels have been documented as disrupting collagen and elastin synthesis in connective tissue, giving rise to abnormalities in the skin, joints, and skeleton. It has been observed in cell cultures that fibrillin-1, the protein altered in Marfan syndrome, experiences reduced levels in cases of Cy deficiency, contributing to the characteristic phenotype exhibited by HCU patients. The Marfanoid syndrome, characterized by increased height due to elongated limbs with metaphyseal and epiphyseal overgrowth and a reduced upper-to-lower segment ratio, is common. Arachnodactyly, dry and thin skin, and brittle hair may also be present. Nearly constant osteoporosis predisposes individuals to conditions such as scoliosis, pathological fractures, and vertebral collapse. Other common deformities encompass genu valgum, pectus excavatum or carinatum, and cavus feet. Joint mobility limitation, especially in the limbs, often contrasts with the joint laxity seen in Marfan syndrome.

A certain degree of intellectual disability is prevalent, likely due to the competitive inhibition of amino acid transport through the blood–brain barrier and the challenges in neurotransmitter synthesis arising from high concentrations of Met and Hcy. Behavioral and personality disorders are also frequently observed [9,20,45]. see Table 2.

### 5.4. Neuropsychiatric Abnormalities

Neurological manifestations include psychomotor delays, psychiatric disorders, and seizures. Psychiatric symptoms affect up to 50% of HCU patients, predominantly manifesting as mood disorders, anxiety, and obsessive–compulsive disorder [46]. Instances of psychotic symptoms have also been reported, and these can occur even in the absence of other clinical symptoms. Some HCU patients with no history of psychiatric issues or risk factors have exhibited visual hallucinations, agitation, and a poor response to antipsychotic medications [47,48]. Mental disability is prevalent and can be linked with other symptoms such as thromboembolic complications, lens dislocation, or a Marfanoid appearance [49]. See Table 2.

### 5.5. Kidney Disease

HHcy has notable implications for renal health, necessitating consideration in cases of atypical hemolytic uremic syndrome (aHUS) and unexplained renal function decline with associated symptoms. In adults, cblC deficiency may present with proteinuria, hypertension, chronic kidney disease (CKD), and aHUS. Routine genetic panels for aHUS and chronic renal disorders should include genes from the intracellular cobalamin pathway [50].

Chronic kidney disease (CKD) emerges as a common acquired cause of HHcy across age groups. Hcy levels in CKD patients are significantly elevated, with a prevalence ranging from 85% to 100%. There is a positive correlation between creatinine levels and Hcy concentrations, emphasizing the link between Hcy and the degree of kidney disease. The pathogenic mechanism involves altered Hcy metabolism rather than reduced excretion. In cblC deficiency, early treatment with hydroxocobalamin and folates may potentially reverse renal insufficiency [51,52].

HHcy contributes to thrombotic microangiopathy (TMA), leading to HUS. Although rare, the association with HUS has been observed in newborns with cblC deficiency. Elevated plasma levels of Hcy and MMA play a role in TMA pathogenesis, disrupting antithrombotic properties of vascular endothelium and promoting vascular thrombosis. Hcy-thiolactone and MMA induce cellular damage, impacting renal cells. Tubulointerstitial nephritis and proximal renal tubular acidosis have been reported in cblC remethylation defects [53]. See Table 2.

Renal infarction is an infrequent but serious consequence of HHcy, contributing to multiple thromboembolic events affecting various circulations. Patients with HCU may experience arterial hypertension due to renal artery thrombosis. Elevated Hcy levels contribute to atherosclerosis and thrombosis through various mechanisms, including LDL oxidation, endothelial growth inhibition, smooth cell proliferation stimulation, and interference with coagulation and fibrinolysis. Mild to moderate increases in Hcy may also serve as markers of tissue damage or repair, with the association between HHcy and vascular disease becoming more pronounced after a vascular event [53].

Understanding the multifaceted impact of HHcy on renal health is crucial for comprehensive patient care. The routine inclusion of Hcy assessment in relevant genetic panels can aid in early detection and intervention, potentially mitigating severe complications associated with HHcy.

### 5.6. Ocular Abnormalities

Metabolic disorders affecting Hcy metabolism can have significant implications for various components of the eye, presenting challenges and opportunities in the field of ophthalmology. Ophthalmologists encounter two primary clinical scenarios: patients with elevated Hcy as part of a known metabolic disorder, where ocular abnormalities manifest because of the underlying condition, and those presenting with ocular issues, prompting suspicions of an underlying metabolic disorder. Ophthalmologists must be well versed in the ocular manifestations associated with HHcy for diagnostic purposes and ongoing disease monitoring [54]. See Table 2.

In classic HCU, a hallmark feature is inferior lens subluxation, with *ectopia lentis* being a consistent clinical manifestation. Approximately 85% of non-responders to vitamin B6 experience lens dislocation before the age of 12, often bilaterally and located in the lower or nasal regions. This condition can lead to complications like retinal detachment, strabismus, severe myopic astigmatism, and cataracts, typically emerging around the age of 15 [17].

MTHFR deficiency, among other remethylation disorders, is associated with thrombotic events like central retinal vein occlusion (CRVO). Evaluation of Hcy levels is recommended in CRVO patients, particularly in the absence of typical vascular risk factors, those under 55 years of age, or instances of bilateral involvement [55,56].

CblC deficiency, homocystinuria, presents a diverse ocular phenotype and is notable for its association with childhood maculopathy. Early-onset cases exhibit progressive retinal conditions, ranging from subtle retinal nerve fiber layer loss to advanced optic and macular atrophy with characteristic “bone spicule” pigmentation, accompanied by nystagmus, abnormal vision, and strabismus. Late-onset cases do not consistently show evidence of retinal degeneration or optic atrophy [57,58,59]. Understanding these ocular manifestations is pivotal for ophthalmologists in providing comprehensive care and improving patient outcomes in the context of metabolic disorders affecting Hcy metabolism.

### 5.7. Hearing Loss

Sensorineural hearing loss (SNHL), although multifactorial, has been associated with some rare diseases involving alterations in HHcy. This is the case of the combined MMA and HCU type (i.e., CblC), which results in a decreased production of cofactors for mutase (adenosylcobalamin) and methionine synthase (methylcobalamin) and, in turn methylmalonyl-CoA, with elevated Hcy levels in the cerebrospinal fluid that correlate with unilateral SNHL [60,61,62]. There is also an age-dependent strain-specific expression of methionine cycle genes in the mouse cochlea and further regulation during the response to noise damage. Thus, some remethylation disturbances (betaine-homocysteine S-methyltransferase deficiency) cause increased susceptibility to noise-induced hearing loss associated with plasma HHcy [63]. See Table 2.

### 5.8. Reproductive Medicine and Pregnancy

Elevated Hcy levels during pregnancy can adversely affect implantation, embryonic development, fetal growth, and maternal health, increasing the risk of conditions such as preeclampsia [64]. This is particularly significant in women with a history of pregnancies complicated by elevated Hcy as it serves as a crucial risk indicator, impacting vascular function during pregnancy and heightening the risk of adverse effects on embryonic development [65]. Pregnant women predisposed to thrombosis face an additional risk, with elevated Hcy potentially disrupting proper embryonic development. Folate deficiency is recognized as a contributor to an increased risk of neural tube defects in the developing embryo [66]. See Table 2.

The measurement of plasma Hcy levels provides an indirect means of assessing folate deficiency, emphasizing the importance of monitoring Hcy as a potential marker for folate status during pregnancy. Moreover, moderately elevated Hcy levels before conception are associated with lower performance on neurodevelopmental tests at four months and cognitive assessments at six years [67]. Understanding and addressing these relationships are crucial for maternal and fetal health during pregnancy.

## 6. Management of Patients with Suspected HHcy

### 6.1. Evaluation

When encountering arterial or venous thrombotic vascular events, the possibility of moderate to severe HHcy should be considered. The evaluation process begins with a thorough medical history and physical examination to identify signs consistent with HCU. Severe HHcy-related disorders may manifest as developmental delays or behavioral issues in children, while adults may present with vascular disease along with hematological, neuropsychiatric, ocular, or renal disorders. In women with high-risk pregnancies or infertility, HHcy should be considered. The presence of classic HCU is considered when certain features, such as a Marfanoid morphotype, lens ectopia, severe non-familial myopia, skeletal deformities, arterial or venous thrombotic vascular events, intellectual disability, and psychiatric symptoms, are identified. Kidney disease and infertility are also associated conditions.

For pediatric and young adult patients, HHcy exclusion is crucial in those displaying a Marfanoid habitus or compatible dysmorphic features, particularly if accompanied by intellectual disabilities. In young adults, ruling out HHcy is recommended in cases of thromboembolic disease (especially under 55 years without apparent causes), recurrent or unusual thromboses, and patients with peripheral embolisms, early coronary disease, or pulmonary hypertension linked to chronic venous thromboembolic disease.

When moderate or severe HHcy is suspected (Hcy levels > 31 μmol/L), comprehensive testing is recommended, including measurements of plasma Hcy, blood amino acids, vitamin B12, and folic acid levels. A study of organic acids in urine to determine methylmalonic acid (MMA) levels is also advised. The diagnostic possibilities are outlined in an algorithm (Figure 3) and summarized in Table 3, facilitating a systematic approach to confirm or rule out HHcy.

### 6.2. Treatment

The treatment approaches for different forms of HHcy vary, reflecting the diverse underlying causes and associated complications. In general, management options include vitamin B6 for susceptible forms or vitamin supplementation and a low-protein diet with betaine administration.

For mild HHcy, studies examining the impact of folic acid supplementation on cardiovascular and thromboembolic risk have yielded mixed results. The American Heart Association suggests that folic acid supplementation may reduce Hcy levels, but the reduction in cardiovascular risk is uncertain. However, treatment aiming to lower Hcy levels has shown promise in slowing carotid atherosclerosis progression, aiding primary stroke prevention, and delaying brain atrophy in mild cognitive impairment. The supplementation of vitamin B6, B12, or folic acid, considering the balance of risks and benefits, appears generally advantageous.

Classic HCU necessitates a multifaceted approach. The initial treatment involves a therapeutic trial to determine vitamin B6 sensitivity, with one-third of patients responding positively. A low-protein diet, essential amino acid supplementation devoid of Met, and folate and vitamin B12 supplementation support Hcy remethylation via MS. Betaine supplementation, promoting remethylation through alternative pathways, is considered. The optimal dosage of betaine can vary, but general recommendations for betaine supplementation often range from 500 mg to 3 g per day.

In cases resistant to vitamin B6, strict low-protein diets, Met-free essential amino acid formulas, and betaine administration are options. The recommended dose of betaine in children and adults is 100mg/kg/day divided into two doses per day, whereas in some patients, doses above 200 mg/kg/day are needed to reach therapeutic goals. The primary treatment goal is to maintain total Hcy concentrations below 50 μmol/L, which is crucial for preventing vascular events or stabilizing neurological and bone involvement. However, the effectiveness in managing ocular involvement remains limited.

HCU due to remethylation disorders, exemplified by CblC deficiency, involves a comprehensive treatment approach. Hydroxocobalamin (OHCbl) is preferred over cyanocobalamin, administered parenterally to reach optimal vitamin B12 blood levels. Betaine doses are adjusted to optimize Hcy and Met levels. Unlike HCU, remethylation disorders do not necessitate protein restriction, emphasizing the importance of maintaining normal Met levels. Folic acid and carnitine supplementation have shown limited utility in treating remethylation disorders.

The varied treatment strategies underscore the complexity of HHcy management, requiring tailored approaches based on the specific form and its associated complications. Individualized treatment plans, incorporating dietary modifications and targeted supplementation, are essential for optimizing outcomes in patients with HHcy.

## 7. Hypoprotein Diet

Developing a hypoprotein diet involves meticulous consideration of food types, protein content, and overall nutritional value. The goal is to ensure adequate energy intake while controlling protein levels. The diet includes hypoprotein foods, amino acid mixtures without methionine, and supplements of minerals, vitamins, and trace elements. The principles of developing and implementing such a diet are crucial for patients managing metabolic disorders.

The hypoprotein diet categorizes foods based on Met or protein content into prohibited, controlled, and freely allowed groups. Prohibited foods, high in protein, include animal-origin foods like meat, fish, eggs, and certain plant-based foods such as legumes and nuts. Controlled foods, providing essential amino acids, are consumed in limited quantities based on individual tolerance. Freely allowed foods either are protein-free or have negligible protein content and include fats, sugary products, certain flours, seasonings, and specialized hypoprotein foods.

The system of weighted rations aids in recipe formulation, is useful for preventing the exceeding of daily protein rations. It involves selecting an arbitrary Met unit, simplifying the creation of varied menus without risking errors. Patient education focuses on understanding authorized foods, weight equivalences, and the system of weighted rations, ensuring proper meal planning.

Hypoprotein products, including bread and biscuits, are crucial for energy intake. These artificial foods, with reduced protein content, substitute traditional starchy items. Amino acid mixtures, devoid of methionine, are essential to prevent amino acid deficiencies in strict diets. Administered at least twice daily, they contribute to metabolic balance, albeit with potential taste challenges.

In addition to vitamins like pyridoxine (B6) and folic acid, minerals, vitamins, and trace elements are necessary to meet the recommended intake levels. Commercially prepared products can supplement these requirements when amino acid supplements are included, streamlining daily administration.

Overall, adherence to the principles of a hypoprotein diet is vital for patients with IEM. Categorizing foods, understanding weighted rations, and incorporating hypoprotein products, amino acid mixtures, and supplements contribute to effective management and improved patient outcomes. Education plays a key role in empowering patients to make informed dietary choices, ensuring the successful implementation of a hypoprotein diet tailored to their needs.

## 8. Conclusions

Based on clinical experience and a review of the literature, it is recommended to measure plasma Hcy levels in any situations where treatable HCU is a possibility. Adequate evaluation is advised for all patients in whom moderate or severe HHCy (Hcy > 30 μmol/L) is detected.

It is particularly advisable to determine Hcy levels for patients who have experienced arterial or venous thrombotic events, as well as those with mild to moderate cognitive impairment, neuropsychiatric disorders, or associations with ocular or skeletal abnormalities. When a patient exhibits a Marfanoid morphotype, lens ectopia, severe non-familial myopia, skeletal deformities, intellectual disability, and psychiatric symptoms, even if each of these alterations appears in isolation, HCU should be considered. Specific hematological abnormalities, kidney disease, and infertility are clinical situations that may be associated with elevated Hcy levels.

## Figures and Tables

**Figure 1 nutrients-16-00135-f001:**
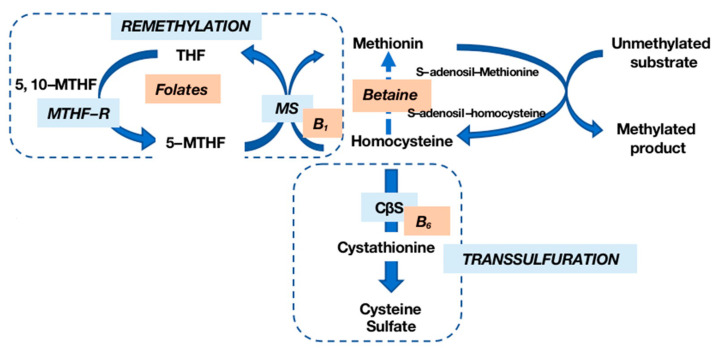
Homocysteine (Hcy) metabolism. Hcy is remethylated into methionine (Met) by methionine synthase (MS) in the presence of vitamin B12 and folates; transsulfuration by cystathionine-β-synthase (CβS), whose cofactor is vitamin B6, allows Hcy to be transformed into cysteine (Cy) and then into sulfate. MTHF-R: methylenetetrahydrofolate reductase.

**Figure 2 nutrients-16-00135-f002:**
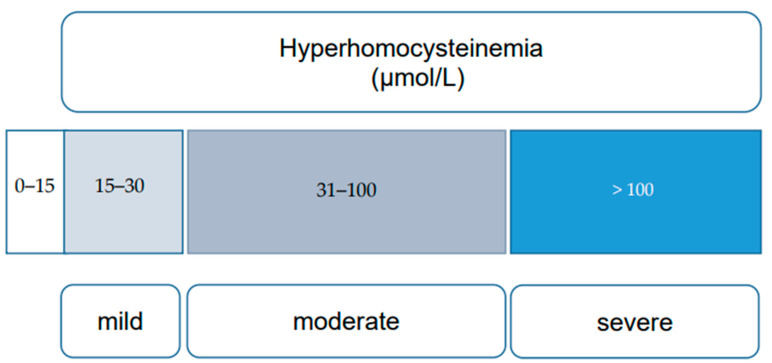
Categories of hyperhomocysteinemia (mild, moderate, and severe), based on plasma homocysteine levels [17].

**Figure 3 nutrients-16-00135-f003:**
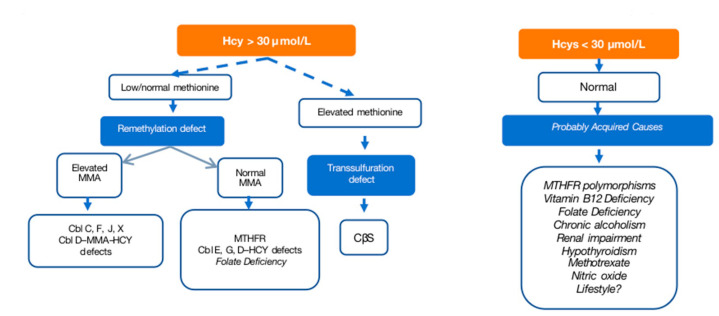
Diagnostic algorithm for hyperhomocysteinemias [17,67] for a cutoff of 30 μmol/L. Cbl: cobalamin; MMA: methylmalonic aciduria; CBS: cystathionine β-synthase deficiency; MTHF-R: methylenetetrahydrofolate reductase.

**Table 1 nutrients-16-00135-t001:** Genetic and acquired causes of hyperhomocysteinemia (HHcy) [21].

Disorders of Hcy Transsulfuration (Decreased Cystathionine-β-Synthase (CβS) Activity)	Disorders of Hcy Remethylation (Altered Methionine Synthase (MS) or Methyltetrahydrofolate Reductase (MTHFR) Activities)
- Genetic deficiency of CβS - Deficiency of vitamin B6 - Drugs (e.g., 6-azauridine)	Genetic deficiency of methyltetrahydrofolate reductase (MTHFR) Defects in Methionine Synthase (MS) Genetic defects in vitamin B12 absorption Disorders of cellular B12 uptake (deficiency of transcobalamin II) Defective lysosomal release of B12 (Cbl-F) Disorders in the conversion of B12 to methyl- or adenosyl-B12 (Cbl-C and Cbl-D) Disorders in the conversion of B12 to methyl-B12 (Cbl-E or Cbl-G) Nutritional deficiency of B12 Nutritional deficiency of folates

**Table 3 nutrients-16-00135-t003:** Basic biochemical alterations in metabolic disorders associated with severe hyperhomocysteinemia (HHcy).

Disorder	Hcy	Met	Cy	MMA	Vit.B_12_	Folates	Macrocytic Anemia
CβS deficiency	↑	N/↑	↓	N	N	N	−
Cbl-C	↑	↓	N/↑	↑	N	N	+/−
Cbl-D/Cbl-F	↑	↓	N/↑	↑	N	N	+
Cbl-E/Cbl-G	↑	↓/N	N/↑	N	N	N	+
MTHFR deficiency	↑	↓	N/↑	N	N	N	+

Hcy: homocysteine; Met: methionine; Cy: cysteine; MMA: methylmalonic acid; Vit.B12: vitamin B12 or cobalamin; MTHFR: methylenetetrahydrofolate reductase. Arrows (↑↓) indicate increase or decrease plama leveles. N normal plasma levels. + presence; − absence.

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
