# Peer review of "Hyperhomocysteinemia in Adult Patients: A Treatable Metabolic Condition"

_nutrients, 2023, doi:10.3390/nu16010135_

Round 1

Reviewer 1 Report

Comments and Suggestions for Authors

Domingo González-Lmuño and colleagues proposed an exhaustive review about hyperhomocysteinemia in adult patients presenting disorders, clinical manifestations, and treatment options. given the complexity of this topic i really appreciated the effort of describing the entire spectrum of metabolic diseases affecting homocysteine and providing a complete overview. as a minor revision I only suggest modifying table 1 distinguish between disorders of hcy transsulfuration (which include dietary deficiency of vitamin b6 as well as cystathionine β-synthase deficiency) and disorders of hcy remethylation (including dietary deficiency of folate or vitamin B12, defects in MTHFR, defects in methionine synthase and defects in enzymes required for methylcobalamin metabolism).

Author Response

Thank you very much for your comments.

In response to your “minor revision” suggestion, we already modify table 1 according to your suggestions.

Reviewer 2 Report

Comments and Suggestions for Authors

This paper is intended to summarize clinical conditions linked with hyperhomocysteinemia. Style of the manuscript seems superficial,on the texbook level,  not tackling the precise molecular etiopathogenesis, descriptive to clinical signs. Paper misses additional pathogies such as hearing loss or similar pathogies. Figures miss units of Hcy, Line 415 -417 are not understandable. Paper requires deeper insights into the genetic and epigenetic aspects of the pathologies. Also literature data is cited rather  sparse

Comments on the Quality of English Language

English can be improved, the quality is more less acceptable 

Author Response

Thank you for sharing your insights on our manuscript. We appreciate your feedback and would like to address your concerns about the style and depth of our work. A point-by-point response to your comments is included "Please see the attachment." English revised version (with the corresponding certificate) is included
